# Applications of Information Theory Methods for Evolutionary Optimization of Chemical Computers

**DOI:** 10.3390/e22030313

**Published:** 2020-03-10

**Authors:** Jerzy Gorecki

**Affiliations:** Institute of Physical Chemistry, Polish Academy of Sciences, Kasprzaka 44/52, 01-224 Warsaw, Poland; jgorecki@ichf.edu.pl

**Keywords:** chemical computing, oscillatory reaction, genetic optimization, classification problem, interacting oscillators

## Abstract

It is commonly believed that information processing in living organisms is based on chemical reactions. However, the human achievements in constructing chemical information processing devices demonstrate that it is difficult to design such devices using the bottom-up strategy. Here I discuss the alternative top-down design of a network of chemical oscillators that performs a selected computing task. As an example, I consider a simple network of interacting chemical oscillators that operates as a comparator of two real numbers. The information on which of the two numbers is larger is coded in the number of excitations observed on oscillators forming the network. The parameters of the network are optimized to perform this function with the maximum accuracy. I discuss how information theory methods can be applied to obtain the optimum computing structure.

## 1. Introduction

The domination of semiconductor technology in modern information processing comes from the fact that man-made semiconductor logic gates are reliable, fast and inexpensive. Their mean time of trouble-free work is probably longer than a human life time. The technology allows combining the logic gates together, leading to complex information processing devices. Therefore, the dominant information coding is based on the binary representation and the design uses the bottom-up strategy, allowing to make more complex information processing devices as a combination of the simple ones [1]. Such an approach is highly successful, leading to exaFLOPS calculations, cloud computing or the fast internet.

Information processing in living organisms is based on chemical reactions [2]. Although, in the case of numerical calculations, the chemical information processing is many orders of magnitude slower than that achieved in silicon circuits, the dedicated chemicals computers (read brains) can execute many algorithms faster than a powerful modern electronic computer. This remark applies to problems that require three-dimensional reconstruction of space and mapping complex motion. Most humans can learn to drive a car, whereas the algorithms for autonomous cars are still far from being applicable. Chemical information processing in living organisms is usually reliable during an animal life. However, the activity time of chemical information processing media used in experiments, as for example, the Belousov–Zhabotinsky (BZ) reaction, is measured in hours rather than in years. Therefore, the bottom-up approach to the construction of chemical information processing devices is not useful for potential experimental verification of theoretical concepts because of the short lifetime of the components.

There are many strategies in which a chemical medium can be applied for information processing. Information can be coded in concentrations of reagents, in the spatial structures or in the spatio-temporal evolution. An excitable chemical medium allows for easy realization of logic gates [3,4,5,6,7]. In such gates, the input and output states are coded in the presence or absence of an excitation at a selected point of the computing medium within a specific time interval. However, the most effective algorithms are obtained if the chemical information processing medium works in parallel. For example, it happens if reactions at different points are coupled by the diffusion. Two classical algorithms of reaction–diffusion computing belong to such class. One of them is the prairie-fire algorithm, which allows finding the shortest path in the labyrinth using wave propagation in an excitable medium [8,9]. The other is the Kuhnert algorithm for image processing with the light inhibited variant of the oscillatory BZ-reaction [10,11,12].

The BZ-reaction has probably been the most studied chemical reaction where the nonlinear phenomena are clearly manifested [13,14]. The reaction is an oxidation of an organic substrate by bromine compounds in an acidic environment and in the presence of a catalyst. The BZ-reaction became famous because oscillations can be easily observed, as the changes in concentrations of the catalyst in different oxidation forms are reflected by the medium color. If the ferroine is used as the catalyst then the medium is red when the reduced catalyst ($Fe{\left(phen\right)}_{3}^{2+}$) is dominant and the medium becomes blue for a high concentration of the catalyst in the oxidized form ($Fe{\left(phen\right)}_{3}^{3+}$). The reaction includes an autocatalytic production of the reaction activator ($HBrO_{2}$). If the medium is spatially distributed and if the diffusion of the activator is allowed, then the region corresponding to a high concentration of the activator can trigger the reaction around and a pulse of the activator propagating in space can appear. The interest in a BZ-reaction as a medium for chemical information processing comes from the fact that its properties are similar to those observed for the nerve system [2]. Using a spatially distributed medium, one can form channels where propagation of excitation pulses is observed. These pulses interact (annihilate) one with another and can change their frequency on non-excitable junctions between channels [15]. The output information is usually coded in the presence of an excitation pulse at a given point and at a specific time. The successes of reaction–diffusion computing with an excitable medium seem to confirm the key role of excitability in biological information processing [16]. Perhaps this is true, but the recent results suggest that an oscillatory medium and information coded in the number of oscillations can also be efficiently applied for chemical computing.

If the ruthenium complex (*Ru*(*bpy*)_3_) is used as the catalyst, then the BZ reaction is photosensitive and illumination with the blue light produces $Br^{-}$ ions that inhibit the reaction [17,18,19]. After illumination of such an oscillatory medium, excitations are rapidly damped and the system reaches a stable, steady state. On the other hand, the oscillatory behavior re-appears immediately after the illumination is switched off [20]. The existence of external control is very important for information processing applications because it allows inputting information into the computing medium [10,11,12]. For the analysis presented below, it is sufficient to assume that the controlling factor has an inhibiting effect. In the following part of the paper, following the analogy with the photosensitive BZ-reaction, I use the word illumination to describe the control factor.

Our recent results suggest that reasonably accurate database classifiers can be constructed with a network of coupled chemical oscillators [21,22,23]. The database records are assumed to have a form of predictors followed by the record type. The classification algorithm is supposed to return a correct record type if the predictor values are known. In this approach, it is assumed that each oscillator in the network can be individually inhibited by an external factor. We can use this factor to introduce the input information and to control the evolution of the medium. Oscillators in the considered information processing networks belong to two types. There are input oscillators and their illuminations are related to predictor values of a given record. There are also so called ”normal” oscillators and their illuminations are fixed. The normal oscillators are supposed to moderate interactions in the medium and optimize them for a specific problem. Therefore, the locations of normal oscillators and their illuminations define the program executed by the network. It is also assumed that the output information about the record type can be extracted from the number of excitations (the number of maxima of a specific reactant) observed at a selected set of oscillators within a fixed interval of time. In such an approach, information processing is a transient phenomenon. It does not matter if the system approaches a stationary state over a long time or not.

It has been demonstrated [21,22,23] that the top-down approach can be successfully applied to design classifiers based on coupled chemical oscillators. Within the top-down approach, we first specify the function that should be performed by the considered system. Next, we search for possible factors that can modify the system evolution and increase its information processing ability. Finally, we combine all these factors and apply them to achieve the optimum performance. The top-down design reflects the idea of Evolution, where the struggle to survive is the goal, but the measures to achieve this goal and possible synergy between different factors increasing the fitness of an individual organism are not fully understood. Having in mind that the structure of the brain is a product of evolution, it is not surprising that the evolutionary optimization can be a useful tool for finding the values of parameters describing a computing medium that performs a specific operation. The evolutionary optimization [24,25] works on a population of classifiers. The best classifiers are allowed to recombine their parameters and to produce an offspring that is included in the next generation. Spontaneous mutations of classifier parameters are also taken into account. As a result of trial and error, the fitness of the best classifier increases with the number of generations [21,22,23]. Of course, it can happen that the considered medium is completely useless for the task we like to perform. Nevertheless, even in cases where the selected medium is needless for a solution of the considered information processing task, the application of evolutionary optimization should allow us to estimate the medium usefulness.

In the following, I investigate the classifier corresponding to the problem of which of the two real numbers x,y∈[0,1] is larger as an example application of information theory to chemical computing. The methods of information theory suggest the type of fitness function for the evolutionary optimization of a classifier formed by coupled chemical oscillators. The motivation for the research is to illustrate that chemical computing can be efficient if we optimize the computing medium and give some margin for potential errors. Actually, this is something that characterizes information processing in the living organisms (“errare humanum est”). For the verification, if y>x, a network of oscillators that solves this problem with reasonable accuracy can be quite simple, and as shown below, it can be made of just three oscillators.

## 2. Results

The problem of which of the two numbers x,y∈[0,1] is larger has a direct geometrical interpretation. A pair (x,y) represents a point in the unit square. The problem if y>x is equivalent to determining if the corresponding point is located above the unit square diagonal (cf. Figure 1a).

Chemistry suggests many strategies that allow us to verify which of two numbers is larger. For example, on can consider a reaction:X+Y→products
in which one molecule of X is consumed together with a single molecule of Y. One can start such a reaction with initial concentrations of [X] and [Y] equal to x and y, respectively. After a long time, the only remaining molecules are from the reagent that was the majority: there are the molecules of X if x>y and molecules of Y if the reverse relation was held. So spectroscopy should give us an answer to which of the two numbers x,y is larger.

The same strategy applies if we use a more complex reaction, as for example:aX+bY+Z→products
but in such a case, we obtain the relationship between the values of ax and by.

The problem of the relationship between two numbers x,y can also be solved with an oscillatory reaction that is inhibited by light. It is known that the period of the photosensitive BZ-reaction increases with illumination [26]. The numbers x,y can be translated into light intensities I(x) and I(y) using the function I(q)=α∗q+β(α,β>0) such that for q∈[0,1] the values of I(q) are in the range corresponding to rapid changes in the period. Next, we need to apply illuminations I(x) and I(y) to two identical reactors and measure the periods. The oscillator characterized by a longer period was illuminated by light with intensity corresponding to the larger of two numbers x,y.

Here, I consider yet another strategy of solving the problem of which of the two numbers is larger by formulating it as a database classification. Problems of database classification are quite general because many algorithms can be re-formulated as database classifications. Let us consider an algorithm *A* that returns a discrete output on an input in the form of *N* real numbers belonging to the algorithm domain $\left(D_{A}\right)$. Moreover, we assume that the number of possible answers is finite. Let (K+1)∈N describe the number of answers. Formally, we describe such an algorithm as a map:(1)$$A:R^{N}⊃D_{A}\to O,$$
where O={0,1,…,K}. If we consider an element $\left(p_{1}\ldots ,p_{N}\right)\in \;D_{A}$, then:(2)$$A:D_{A}∋p=\left(p_{1}\ldots ,p_{N}\right)\mapsto t_{p}\in \left\{0,1\ldots ,K\right\}.$$

Let us introduce a set $E_{A}\left(L\right)$, which contains *L* arguments of the algorithm *A*:(3)$$E_{A}\left(L\right)=\left\{p^{s}=\left(p_{1}^{s},p_{2}^{s},\ldots ,p_{N}^{s}\right);p^{s}\in D_{A}\wedge s\in \left\{1,2,\ldots ,L\right\}\right\}.$$

We can generate the database $F_{A}\left(L\right)$ made of records constructed in the following way:(4)$$F_{A}\left(L\right)=\left\{\left(p_{1}^{s},p_{2}^{s},\ldots ,p_{N}^{s},t_{p^{s}}\right);\left(p_{1}^{s},p_{2}^{s},\ldots ,p_{N}^{s}\right)\in E_{A}\left(L\right)\wedge t_{p^{s}}=A\left(p^{s}\right)\right\}.$$

Each record of $F_{A}\left(L\right)$ contains *N* predictors $p_{1}^{s},p_{2}^{s},\ldots ,p_{N}^{s}$ and the record type $t_{p^{s}}$. The classifier of the database $F_{A}\left(L\right)$ is supposed to return the correct record type if the predictor values are used as the input. A classifier that correctly classifies any dataset $F_{A}\left(L\right)$ can be seen as a program that executes the algorithm *A*. For example, the XOR operation
XOR:{0,1}×{0,1}→{0,1}
can be completely described by the classification of database:$$F_{XOR}\left(4\right)=\left\{\left(0,0,0\right),\left(0,1,1\right),\left(1,0,1\right),\left(1,1,0\right)\right\}.$$
Therefore, the chemistry based classifier of $F_{XOR}\left(4\right)$ is a chemical realization of the XOR gate. The same approach applies to any logical operations involving those of multivariable ones.

Within the formalism presented above, the problem which of two numbers is larger can be seen as an algorithm $A_{>}$:$$A_{>}:\left[0,1\right]\times \left[0,1\right]∋\left(x,y\right)\mapsto t_{\left(x,y\right)}\in \left\{0,1\right\},$$
where $t_{\left(x,y\right)}=1$ if x≥y and $t_{\left(x,y\right)}=0$ iff x<y. Therefore, the problem can be formulated as the classification problem for databases in the form:$$F_{>}\left(L\right)=\left\{\left(x^{s},y^{s},t_{\left(x^{s},y^{s}\right)}\right);\left(x^{s},y^{s}\right)\in \left[0,1\right]\times \left[0,1\right]\wedge s\in \left\{1,2,\ldots ,L\right\}\right\}$$

I postulate that the problem of $F_{>}\left(L\right)$ database classification can be solved by a network of three coupled oscillators in the geometry illustrated in Figure 1b. I assume that the output information is coded in numbers of excitations observed on oscillators forming the network. Such a method of extracting the output is motivated by the fact that a chemical counter of excitation number can be easily constructed [27], so the information read-out can be done with chemistry.

### 2.1. The Time Evolution Model of an Oscillator Network

In this section, I briefly introduce oscillator networks, discuss the specific properties of the Belousov–Zhabotinsky reaction that can be useful for construction of classifiers and introduce a simple model of network time evolution. The detailed information can be found in [28].

The networks of chemical oscillators can be formed in different ways. One can use individual continuously stirred chemical reactors (CSTRs) and link them by pumps ensuring the flow of reagents [29,30]. Alternatively, networks of oscillators can be formed by touching droplets containing a water solution of reagents of an oscillatory BZ reaction stabilized by lipids dissolved in the surrounding oil phase. If phospholipids (asolectin) are used, then BZ droplets communicate mainly via an exchange of the reaction activator ($HBrO_{2}$ molecules) that can diffuse through the lipid bilayers and transmits excitation between droplets [31]. A high uniformity of droplets that form the network can be achieved if droplets are generated in a microfluidic device [32]. One can also use DOVEX beads or silica balls [33] to immobilize the catalyst and inhibit oscillations by illumination or electric potential [34].

The simplest mathematical model of a BZ-reaction describes the process as an interplay between two reagents: the activator and the inhibitor (the oxidized form of the catalyst). Two variable models, such as the Oregonator [35] or Rovinsky–Zhabotinsky model [36], give a pretty realistic description of simple oscillations, excitability and the simplest spatio-temporal phenomena. However, the numerical complexity of models based on kinetic equations is still substantial, and they are too slow for large scale evolutionary optimization of a classifier made as a network of oscillators. Here, following [28], I use the event based model. I assume that three different phases: excited, refractive and responsive can be identified during a single oscillation cycle of a typical chemical oscillator [21,22,28]. The excited phase denotes the peak of activator concentration. A chemical oscillator in the excited phase is able to spread out the activator molecules and trigger excitations in the medium around. In the refractory phase, the concentration of the inhibitor is high, and an oscillator in this phase does not respond to an activator transported from oscillators around. In the responsive phase, the inhibitor concentration decreases. An oscillator in the response phase can get excited by interactions with an oscillator in the excited phase. Following the previous papers [28], the oscillation cycle combines the excitation phase lasting 1 s, the refractive phase, lasting 10 s and the responsive phase that is 19 s long (cf. Figure 2). For an isolated oscillator, the excitation phase appears again after the responsive phase ends and the cycle repeats. Thus, the period of the cycle is 30 s. Oscillations with such a period have been observed in experiments with BZ-medium [20]. The separation of oscillation cycle into phases allows introducing a simple model for interactions between individual oscillators. An oscillator in the refractory phase does not respond to the excitations of its nearest neighbors. An oscillator in the responsive phase can be activated by an excited neighbor. I also assume that if an oscillator is in the excitation phase, then 1 s later, all oscillators coupled with it, that are in the responsive phase, switch into the excitation phase. It is also assumed that after illumination is switched on, the phase changes into the refractory one. When the illumination is switched off, the excited phase starts immediately. The model defined above is much faster than models based on the kinetic equations.

### 2.2. The Computing Medium Made of Interacting Oscillators and Its Evolutionary Optimization

In this section, I introduce the parameters needed for numerical simulations of a chemical classifier. In order to use a network of coupled oscillators for information processing, we have to specify the time interval $\left[0,t_{max}\right]$ within which the time evolution of the network is observed. This is an important assumption because it reflects the fact that the information processing is a transient phenomenon. I assume that output information can be extracted by observing the system within the time interval $\left[0,t_{max}\right]$, and it is not important if the network reaches a steady state before $t_{max}$ or not.

It is assumed that the state of each oscillator in the network can be controlled by an external factor (illumination). There are two types of oscillators in the network: the normal ones and the input ones. For a normal oscillator *k*, its activity is inhibited within the time interval $\left[0,t_{illum}^{\left(k\right)}\right]$ ($0\leq t_{illum}^{\left(k\right)}\leq t_{max}$). After the time $t_{illum}^{\left(k\right)}$ the oscillation cycle starts from the excited phase. For a given classification problem the set of times $t_{illum}^{\left(k\right)}$ defining the inhibition of normal oscillators is the same for all processed records. This set of times defines the “program” executed by the network to solve the problem. If an oscillator is considered as an input for the *j*-th predictor and if the predictor value is $p_{j}\in \left[0,1\right]$, then such oscillator is inhibited (illuminated) within the time interval $\left[0,t_{start}+\left(t_{end}-t_{start}\right)*p_{j}\right]$, where the values of $t_{start},t_{end}$ (both $<t_{max}$) are the same for all predictors. The network geometry (i.e., locations of input and normal oscillators and the geometry of interactions between oscillators) together with illuminations of normal oscillators $t_{illum}^{\left(k\right)}$ and times $t_{start}$, $t_{end}$ fully define the network and allow for simulations of its time evolution after the predictor values are selected. In the following, I will assume that the classifier output is represented by the number of excitations observed on oscillators within the time interval $\left[0,t_{max}\right]$.

To verify if the network performs its classification function correctly, I introduce the testing database $F_{>}\left(L=100,000\right)$ composed of records of the form $\left(x,y,t_{\left(x,y\right)}\right)$ where x,y are uniformly distributed random numbers from [0,1]. The classifier accuracy is calculated as the fraction of correct answers for records from $F_{>}\left(L=100,000\right)$. As the zero-th order approximation, one can neglect observation of the classifier evolution and say that *x* is always larger than *y*. If $F_{>}\left(L=100,000\right)$ is selected without bias, then such classifier should show 50% accuracy.

In order to increase the accuracy, we have to optimize the classifier parameters. In the optimization discussed below, they included the maximum time within which the system evolution is observed ($t_{max}$), the locations of input oscillators and the oscillators characterized by fixed illuminations (the normal ones), the times defining the introduction of input values ($t_{start}$ and $t_{end}$) and the illuminations of normal oscillators $t_{illum}^{\left(k\right)}$. Moreover, defining the classifier, we should decide how the output is extracted from the evolution. I select two strategies. I postulate that the classifier output can be read out from the number of excitations on a selected oscillator or as a pair of numbers of excitations observed on two selected oscillators. The quality of a classifier $C_{A}$ corresponding to the algorithm *A* can be estimated if one decides how to read the output information but without interpreting if the obtained result corresponds to the case x≥y or x<y. To do it, we can use the mutual information between the set of record types and the set of classifier outputs. Let us consider a set of arguments $E_{A}\left(L\right)$ (Equation (Equation 3)) and the related database $F_{A}\left(L\right)$ (Equation (Equation 4)) of the algorithm *A*. Let us assume that the classifier $C_{A}$ produces the output string $a_{p^{s}}$ on the record $p^{s}\in E_{A}\left(L\right)$:$$C_{A}:E_{A}\left(L\right)∋p^{s}\mapsto a_{p^{s}}$$

Now let us consider three lists:$$B=\{a_{p^{s}};p^{s}\in E_{A}\left(L\right)\}$$
$$O=\{t_{p^{s}};\left(p^{s},t_{p^{s}}\right)\in F_{A}\left(L\right)\}$$
and
$$OB=\{\left(t_{p^{s}},a_{p^{s}}\right);\left(p^{s},t_{p^{s}}\right)\in F_{A}\left(L\right)\}$$

The mutual information between *B* and *O* defined as [37]:(5)MI(O,B)=H(O)+H(B)−H(OB)
where H(X) is the Shannon information entropy of the strings that belong to the list *X*. If this list contains *k* different strings then:(6)$$H\left(X\right)=-\sum_{k}r\left(k\right)\log\left(r\left(k\right)\right)$$
where r(k) is the probability of finding the string #k in the list.

In order to calculate MI(O,B), one needs to specify how the output is read out of the network evolution. For example, if the number of excitations observed on a single oscillator within the time interval $\left[0,t_{max}\right]$ is used as the output string, then we should consider all oscillators as potential candidates for the output one. The oscillator, for which the maximum MI(O,B) is achieved, is selected as the output one. In the case of the output coded in a pair of excitation numbers observed on two selected oscillators, one should consider all pairs of oscillators as candidates for the output oscillators. Like in the case of a single oscillator, the pair that produces that maximum MI(O,B) is considered as the output. The maximum value of MI(O,B) obtained within a given method of extracting the output string is considered as the quality (fitness) of the classifier $C_{A}$.

Let us notice that when $a_{p^{s}}=t_{p^{s}}$, so the classifier works without errors, then H(O)=H(B)=H(OB) and MI(O,B)=H(O). On the other hand, if the answers $a_{p^{s}}$ are not correlated with the record types $t_{p^{s}}$ then H(OB)=H(O)+H(B) and MI(O,B)=0. In general, MI(O,B) is an average number of bits of the information about the string $t_{p^{s}}$ we get if we know the string $a_{p^{s}}$. Therefore, we can expect that an increase in MI(O,B) does reflect the rise in classifier accuracy.

To define a classifier, we need to specify: Geom—the network geometry and interactions between oscillators, Loc—location of the input oscillators, $t_{max}$, $t_{end}$, $t_{start}$ and $t_{illum}^{\left(k\right)}$. In the considered problem of verification of which of the two numbers is larger, I assumed that the classifier had a triangular geometry, as illustrated in Figure 1b, and that all oscillators were interconnected. In such a geometry, all oscillators are equivalent. Therefore, we can assume that oscillator #1 is the input of the first predictor. Moreover, the system symmetry allows us to assume that oscillator #2 is the input of the second predictor. Therefore, the only missing element of the classifier structure is the type of oscillator #3 ($Osc_{3}$) and it is a subject of optimization. I did not introduce any constraints on the type of this oscillator. It could be an input oscillator or the normal one.

The optimization of all parameters of the classifier ($Osc_{3},t_{max},t_{end},t_{start},t_{illum}^{\left(3\right)}$) was done using the evolutionary algorithm. The technique has been described in detail in [28]. At the beginning, 1000 classifiers with randomly initialized parameters were generated. Of course, it would be naive to believe that a randomly selected network of oscillators performs an accurate classification of the selected database. The generated networks made the initial population for evolutionary optimization. Next, the fitness (Equation (Equation 5)) of each classifier was evaluated. In order to speed up the algorithm, the fitness was calculated using the database $F_{>}\left(L=10,000\right)$ where predictors were 10,000 randomly generated pairs (x,y). The database $F_{>}\left(L=10,000\right)$ was 10 times smaller than $F_{>}\left(L=100,000\right)$ used to estimate the accuracy of the optimized classifier. However, $F_{>}\left(L=10,000\right)$ is still large enough to contain a representative number of records characterizing the problem. The upper 10% of the most fitting classifiers were copied to the next generation. The remaining 90% of classifiers of the next generation were generated by recombination and mutation processes applied to pairs of classifiers randomly selected from the upper 50% of the most fitting ones. At the beginning, recombination of two parent classifiers produces a single offspring by combining randomly selected parts of their parameters. Next, the mutation operations were applied to the offspring. It was assumed that oscillator #3 can change its type with the probability $p_{type}=0.1$. Moreover, random changes of $t_{max}$, $t_{end}$, $t_{start}$ and $t_{illum}^{\left(k\right)}$ were also allowed. The evolutionary procedure was repeated more than 500 times. A typical progress of optimization is illustrated in Figure 3a and Figure 5a.

### 2.3. Chemical Algorithms for Verification of Which of the Two Numbers Is Larger

In this section, I discuss the optimized networks that verify which of the two real numbers x and y from [0,1] is larger.

At the beginning, let us consider the network that produces the output as a number of excitations observed at a selected node. The progress of optimization on a small database with 1000 records is illustrated in Figure 3a. The optimization continued for 500 evolutionary steps. For the optimized network $t_{max}=130.55$ s. As it has been assumed, oscillators #1 and #2 correspond to inputs of x and y, respectively. The values of $t_{start}$ and $t_{end}$ are 69.54 and 23.79 s, respectively. The highest mutual information was obtained if oscillator #3 was the normal one and $t_{illum}^{\left(3\right)}=49.55$ s. Figure 3b illustrates the numbers of cases corresponding to different numbers of excitations and the relationship between *x* and *y* for records from $F_{>}\left(L=100,000\right)$. The red and green bars correspond to y>x and y<x, respectively. Figure 3b compares the number of cases observed on different oscillators of the network. It is clear that oscillator #3 is useless as the output because it has excited the same number of time by inputs with y>x and by inputs with y<x. On the other hand, for oscillators #1 and #2, the situation is very different. For example, three excitations are observed mainly for y>x, whereas four or five excitations are dominant for y>x. Therefore, we can define the classifier output as the number of excitations on oscillator #1 such that three excitations correspond to y>x and four or five excitations to y≤x. The accuracy of such a classifier is 82.4%. Figure 4 illustrates the location of correctly and incorrectly classified pairs (x,y)∈[0,1]×[0,1] if the output reading rule defined above is used. The correctly classified pairs in which y>x and y<x are marked by red and green points, respectively. Incorrectly classified pairs in which y>x and y<x are marked by blue and black points, respectively. Similar results are obtained when oscillator #2 is used as the output (Figure 3b), but the accuracy is 82.2%. I believe the small difference between using oscillators #1 and #2 as the output is related to randomness in the generated database.

In order to increase the classification accuracy, I considered the same network but another rule of reading out the information. I assumed that the output is coded in a pair of excitation numbers observed on two oscillators. The initial progress of optimization with $F_{>}\left(L=10,000\right)$ is illustrated in Figure 5a. The optimization procedure was continued for 1000 generations. The classifier structure was the same as in the previous case; oscillator #3 was the normal one. The optimized classifier is characterized by $t_{max}=500.00$ s, $t_{start}=498.95$ s, $t_{end}=0.15$ s and $t_{illum}^{\left(3\right)}=71.42$ s. The numbers of excitations observed on oscillators #1 and #2 were used as the classifier answer. The translation of the observed number of excitations into the classifier answer is illustrated in Figure 5b. The mutual information between the classifier outputs and the record types in $F_{>}\left(L=100,000\right)$ is 0.962 and the classification accuracy is 98.2%. Both numbers are remarkably high, especially having in mind that the computing medium is formed by three oscillators only. The optimization procedure has shown that the classifier accuracy strongly depends on $t_{max}$, as illustrated in Figure 5c. In the optimization procedure, the value of $t_{max}$ was limited at 500s. Figure 6 illustrates the location of correctly and incorrectly classified pairs (x,y)∈[0,1]×[0,1] if the output reading rule defined in Figure 5b is used.

### 2.4. Shadows on Optimization Towards the Maximum Mutual Information

As in the previous papers [21,22,23,28], I assumed that the increase in mutual information means that the classifier accuracy increases, so one can use easy-to-calculate mutual information to estimate the quality of a classifier. But does the increase in the mutual information really mean that the classifier accuracy is higher? The example given below illustrates that it is not.

Let us consider the following example. Assume a database with 2N records corresponding to two types *v* and *w* of the form:(7)$$F=\left\{\left(p^{s},v\right);1\leq s\leq N\right\}\cup \left\{\left(p^{s},w\right);N+1\leq s\leq 2N\right\}.$$

Therefore, the list of record types $O=\{t_{p^{s}};\left(p^{s},t_{p^{s}}\right)\in F\}$ contains *N* elements *v* and *N* elements *w* and the Shannon information entropy H(O)=1.

Let us also assume that there is a classifier *C* of the database *F* that produces two outputs $a_{1}$ and $a_{2}$ in the following way: If the record type is *v*, then for *L*(L<N/2) records, the classifier output is $a_{2}$, and for N−L records, the classifier output is $a_{1}$. Let us redistribute these database records such that: $a_{p^{s}}=a_{2}$ if 1≤s≤L and $a_{p^{s}}=a_{1}$ if L+1<s≤N. Moreover, if the record type is *w*, then for *M*(M<N/2) records, the classifier output is $a_{1}$, and for N−M records, the classifier output is $a_{2}$. Yet again let us redistribute these database records such that: $a_{p^{s}}=a_{1}$ if N+1≤s≤N+M and $a_{p^{s}}=a_{2}$ if N+M<s≤2N. Therefore, it is more likely to get the $a_{1}$ answer if the record type is *v* and the $a_{2}$ answer if the record type is *w*. The interpretation: the record type is *v* if the classifier produces $a_{1}$, and the record type is *w* if the classifier produces $a_{2}$ leads to the highest accuracy equal to:(8)Accuracy(k,l)=1−(k+l)/2,
where l=L/N and k=K/N. The contour plot of Accuracy(k,l) is shown in Figure 7a.

The list of classifier answers *B* on the database *F* is made of N−L+M symbols $a_{1}$ and N−M+L symbols $a_{2}$. Therefore;
(9)$$H\left(B\right)\left(k,l\right)=-\left(0.5-k/2+l/2\right)\log_{2}\left(0.5-k/2+l/2\right)-\left(0.5+k/2-l/2\right)\log_{2}\left(0.5+k/2-l/2\right)$$

For the considered classifier, the list OB is formed of *L* strings $\left(v,a_{2}\right)$, N−L strings $\left(v,a_{1}\right)$, *K* strings $\left(w,a_{1}\right)$ and N−K strings $\left(v,a_{2}\right)$. The Shannon information entropy H(OB) equals:(10)$$H\left(OB\right)\left(k,l\right)=-k/2\log_{2}\left(k/2\right)-\left(0.5-k/2\right)\log_{2}\left(0.5-k/2\right)-l/2\log_{2}\left(l/2\right)-\left(0.5-l/2\right)\log_{2}\left(0.5-l/2\right)$$
and the mutual information between record types and answers of the classifier is:(11)MI(k,l)=1+H(B)(k,l)−H(OB)(k,l)

The function MI(k,l) is illustrated in Figure 7b.

Comparing the dependence of Accuracy(k,l) with MI(k,l), we can identify a path in (k,l) space along which one of the functions increases and the other decreases. An example of such a path is marked by the red line in Figure 8a. The values of Accuracy(k,l) and 2∗MI(k,l) along this path are shown in Figure 8b. As seen, the increase in MI(k,l) can slightly decrease Accuracy(k,l).

Having this result in mind, we can say that although optimization based on the mutual information is attractive because it can be easily applied in an optimization program and does not involve direct translation of the evolution into the classification result, a better classifier can still be evolved if its accuracy is used to estimate the classifier fitness.

## 3. Discussion: Why can a Small Network of Chemical Oscillators So Accurately Determine Which of the Two Numbers Is Larger?

The computing networks described above were obtained by a computer program without any human help except of formulating the problem. Now, having the problem solved, we may think over why such high accuracy of classification of which of the two numbers is larger has been achieved.

Let us notice that the problem of which of the two numbers (x,y) is larger can be approximated by solving the problem if x≥0.5. At a first glance, the problems seem quite different. However, if the pairs are randomly distributed in the cube [0,1]×[0,1] (see Figure 1a), then if $x_{1}\geq 0.5$, then with 75% probability, $x_{1}$ is the larger number in a pair $\left(x_{1},y\right)$. And similarly, if $x_{2}<0.5$, then with 75% probability, $x_{2}$ is the smaller number in the pair $\left(x_{2},y\right)$. Construction of a device that checks if a number is smaller or larger than 0.5 using a single chemical oscillator is very easy. Let us assume that the chemical oscillator has a period *T*. Let us consider a classifier made of a single oscillator. This oscillator serves both as the input of *x* and the output. Let us consider the classifier is characterized by $t_{max}=2T-\epsilon$, $t_{start}=t_{max}$ and $t_{end}=0$. For the input value *z*, the input illumination is $t_{input}\left(z\right)=t_{start}+\left(t_{end}-t_{start}\right)*z=\left(2*T-\epsilon \right)*\left(1-z\right)$. If z<1−T/(2∗T−ϵ), then $t_{input}\left(z\right)>T-\epsilon$ so only one excitation of the oscillator is observed in the time interval $\left[t_{input}\left(z\right),t_{max}\right]$. On the other hand, if z≥1−T/(2∗T−ϵ), then $t_{input}\left(z\right)\leq T-\epsilon$ and two excitations are observed. Therefore, such a single oscillator classifier correctly predicts if the input is larger than 0.5 for most of numbers from the interval [0,1]. If we use it to compare two numbers, then using the arguments presented above, we obtain an accuracy of 75%. The idea of a single oscillator classifier presented above can be easily generalized to verify if the input *z* belongs to one of subintervals of [0,1]. Let us assume that $t_{max}=nT-\epsilon$, $t_{start}=t_{max}$ and $t_{end}=0$. For the input value *z*, the input illumination is $t_{input}\left(z\right)=t_{start}+\left(t_{end}-t_{start}\right)*z=\left(n*T-\epsilon \right)*\left(1-z\right)$. Therefore, the oscillator can exhibit 1+floor(z∗n) different numbers of oscillations (assuming the we can neglect small ϵ). Precisely *j* oscillations are observed for $t_{input}\left(z\right)\in \left(\max\left(0,\left(n-j\right)T-\epsilon \right),\left(n-j+1\right)T-\epsilon \right]$, which corresponds to z∈[1−((n−j+1)∗T−ϵ)/(nT−ϵ),1−((n−j)∗T−ϵ)/(nT−ϵ)). Therefore, by observing the number of excitations on a single oscillator, we can estimate the input value with the accuracy ±1/(2n).

Now let us come back to the problem of which of the two numbers in a pair (x,y) is larger. Consider a classifier made of two non-interacting oscillators. One of these oscillators works as an input of *x* and another as an input of *y*, as described above. Next, we compare the number of excitations on both oscillators. If the number of excitations on the oscillator acting as *x* input is larger, then x>y. If the number of excitations is smaller, then y>x (cf. Figure 5b). If the numbers of excitations are equal, then we can assume, for example, x>y because for uniformly distributed data it will give 50% of correct results. Therefore, the accuracy of such a classifier can be estimated as 1−1/(2n). Such a trend can be seen in Figure 5c; thus, the evolutionary algorithm was able to discover the idea of the classifier presented above. For $t_{max}=500$ and T=30, the value of *n* is 17, so we would expect the classifier error around 0.029. The error of the optimized classifier, based on three interacting oscillators, was smaller and equal to 0.018.

## 4. Conclusions and Perspectives

In the paper, I discussed how information theory methods can be applied for optimization of a classifier using a network of interacting oscillators as a chemical medium. The problem has been inspired by the fact that bottom-up design of chemical computers is inefficient. It usually produces large structures requiring a precise fit of reaction parameters [38]. The structures resulting from bottom-up design can be difficult for experimental realization. In this aspect, the methods inspired by information theory provide tools for top-down design of compact and reasonably accurate chemical processing media. The use of mutual information for estimating the quality a of chemical information processing medium seems effective because it does not require specification of the rule used to translate the output information into the classification result. The selection of oscillatory chemical processes for information processing applications comes from the fact that these processes are common and more robust than the acclaimed excitable systems. Chemical oscillations are observed at different spatio-temporal scales going down to nanostructures [39]. Therefore, methods that can help to simplify the design of an information processing medium based on oscillator networks can be important for their future success of chemical computing. It is important that information processing networks of oscillators designed with evolutionary algorithms are not based on the binary logic but use the properties of a medium in an optimized way. The algorithms are parallel just by their design.

As an example, I considered the application of optimization methods based on the maximization of mutual information between the classifier evolution and the record types of the test dataset for finding which of two numbers is larger. The optimized network was able to solve this problem with the accuracy exceeding 98%. Moreover, the evolutionary algorithm lead to a rational strategy that was utilizing the properties of the medium in a rational way. In order to speed-up the simulation procedure, I used the event based model of oscillations and interactions between oscillators, but there are no restrictions on the model. A similar functionality can be expected for oscillating reactions described by chemical kinetic equations. Therefore, I can claim that also in the other cases, the methods of information theory combined with evolutionary optimization are able to reveal the computing potential of the considered medium for the required computing task.

## Figures and Tables

**Figure 1 entropy-22-00313-f001:**
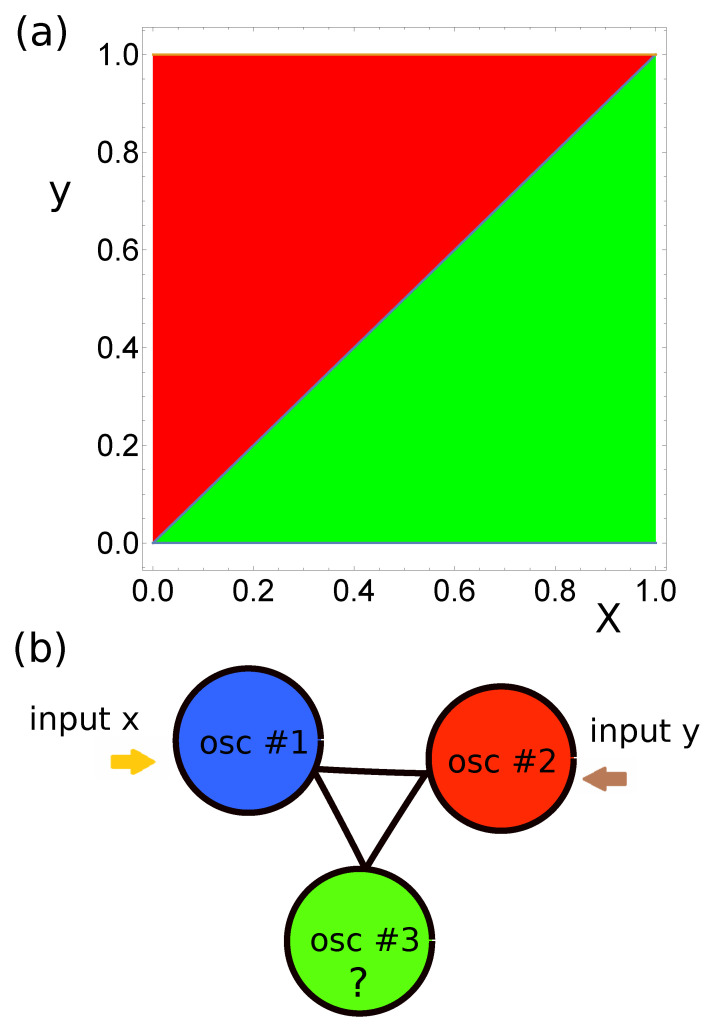
(**a**) The geometrical interpretation of the problem of which of the two numbers is larger. The areas y>x and y<x are colored red and green respectively, (**b**) I postulate that the problem of which of the two numbers is larger can be solved with the illustrated network of three coupled oscillators. Having in mind the symmetry of the problem, I assume that oscillators #1 and #2 are inputs of *x* and *y*. The role of oscillator #3, the network parameters and location of the output oscillators are determined by the evolutionary optimization.

**Figure 2 entropy-22-00313-f002:**
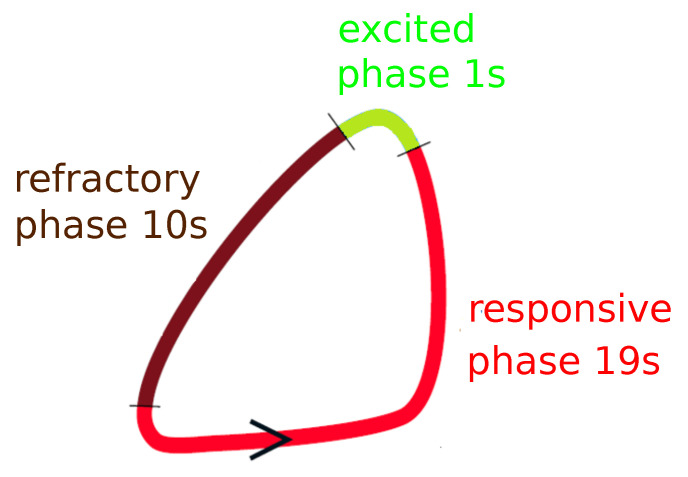
Graphical illustration of the event-based model used to describe the time evolution of an oscillator. Three phases: excited (green), refractory (brown) and responsive (red) follow one after another. The arrow marks the direction of time. Numbers give lengths of corresponding phases.

**Figure 3 entropy-22-00313-f003:**
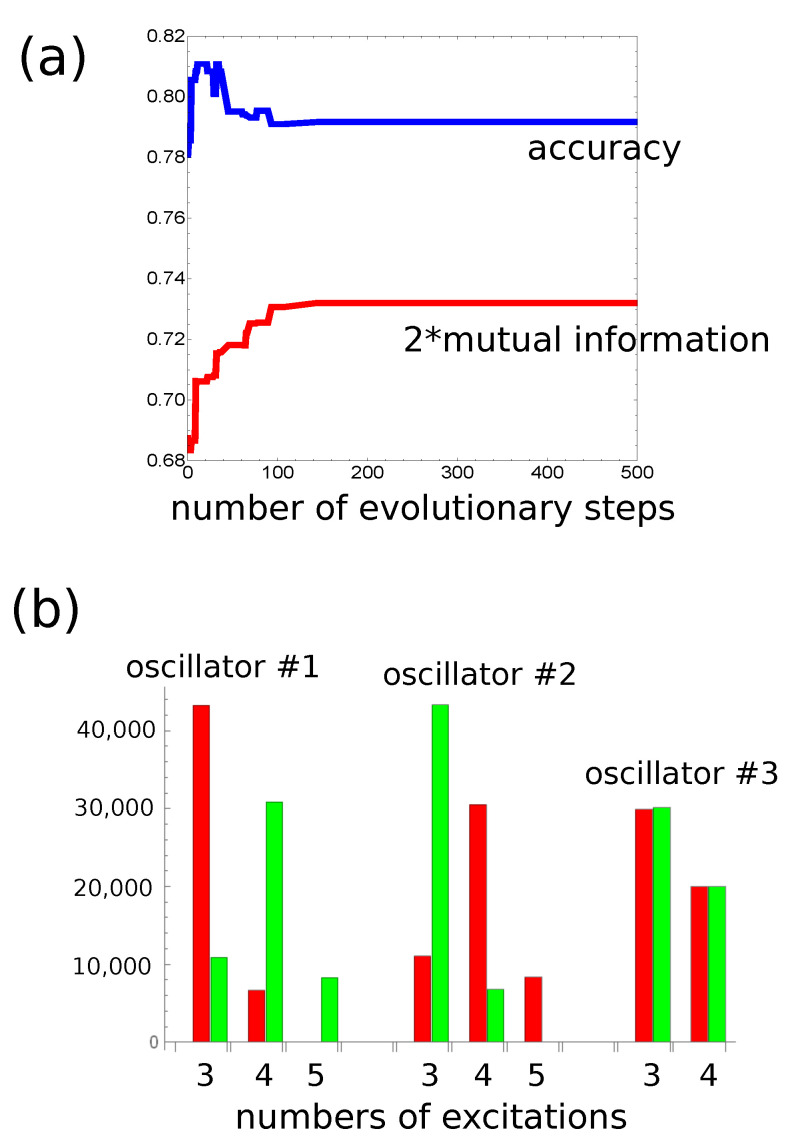
(**a**) The mutual information as a function of an evolutionary step for the classifier in Figure 1b if the number of excitations observed on a single oscillator is used as the output, (**b**) the numbers of cases corresponding to different numbers of excitations and for different relationships between *x* and *y*. The records from $F_{>}\left(L=100,000\right)$ were used. The red and green bars correspond to y>x and y<x, respectively. The number of cases recorded on different oscillators are shown.

**Figure 4 entropy-22-00313-f004:**
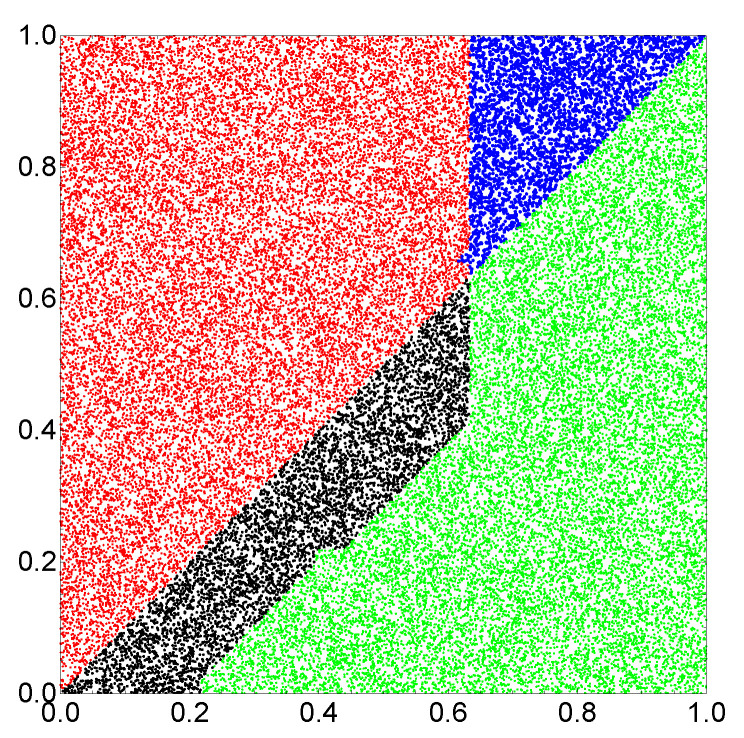
The location of correctly and incorrectly classified pairs (x,y)∈[0,1]×[0,1] if the number of excitations observed on a single oscillator is used as the output. Correctly classified pairs in which y>x and y<x are marked by red and green points, respectively. Incorrectly classified pairs in which y>x and y<x are marked by blue and black points, respectively.

**Figure 5 entropy-22-00313-f005:**
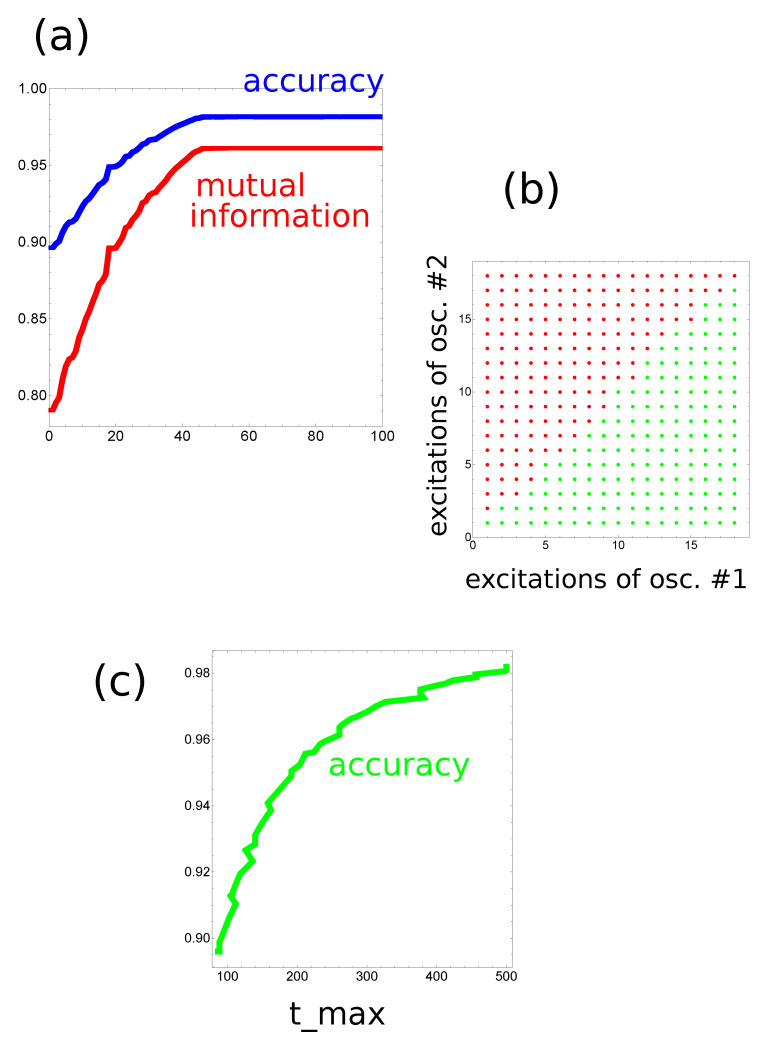
(**a**) The mutual information as the function of an evolutionary step for the classifier in Figure 1b if the pair of excitation numbers observed on two selected oscillators is used as the output. (**b**) The table that translates the numbers of excitations observed on oscillators #1 and #2 into the classifier answer. The red and green points correspond to y>x and y<x, respectively. (**c**) The increase in classifier accuracy as a function of $t_{max}$ observed during the optimization.

**Figure 6 entropy-22-00313-f006:**
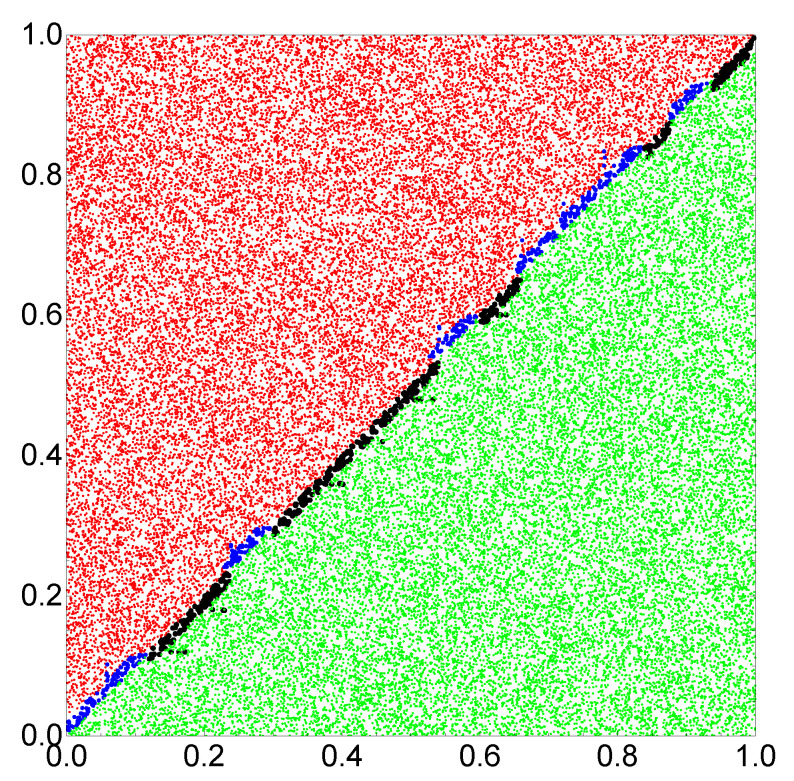
The location of correctly and incorrectly classified pairs (x,y)∈[0,1]×[0,1] if the pair of excitation numbers observed on oscillators #1 and #2 is used as the output. Correctly classified pairs in which y>x and y<x are marked by red and green points, respectively. Incorrectly classified pairs in which y>x and y<x are marked by blue and black points, respectively.

**Figure 7 entropy-22-00313-f007:**
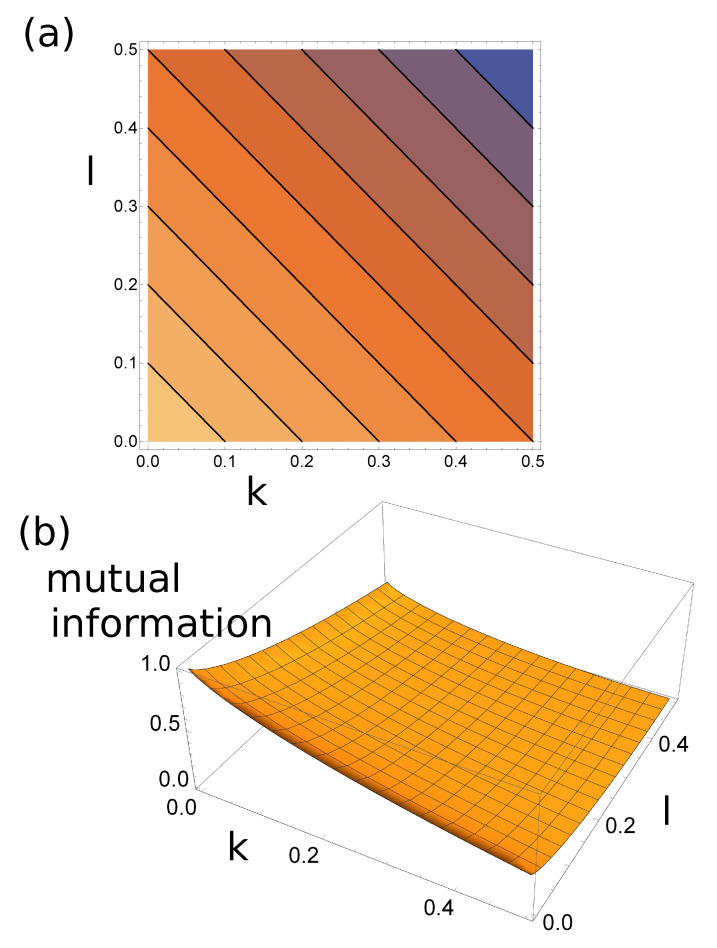
(**a**) The countour plot of Accuracy(k,l) and (**b**) the mutual information MI(k,l) for the classifier discussed in Section 2.4 as functions of *k* and *l*. The contour lines in (a) are equally distributed between 0.5 and 1.0.

**Figure 8 entropy-22-00313-f008:**
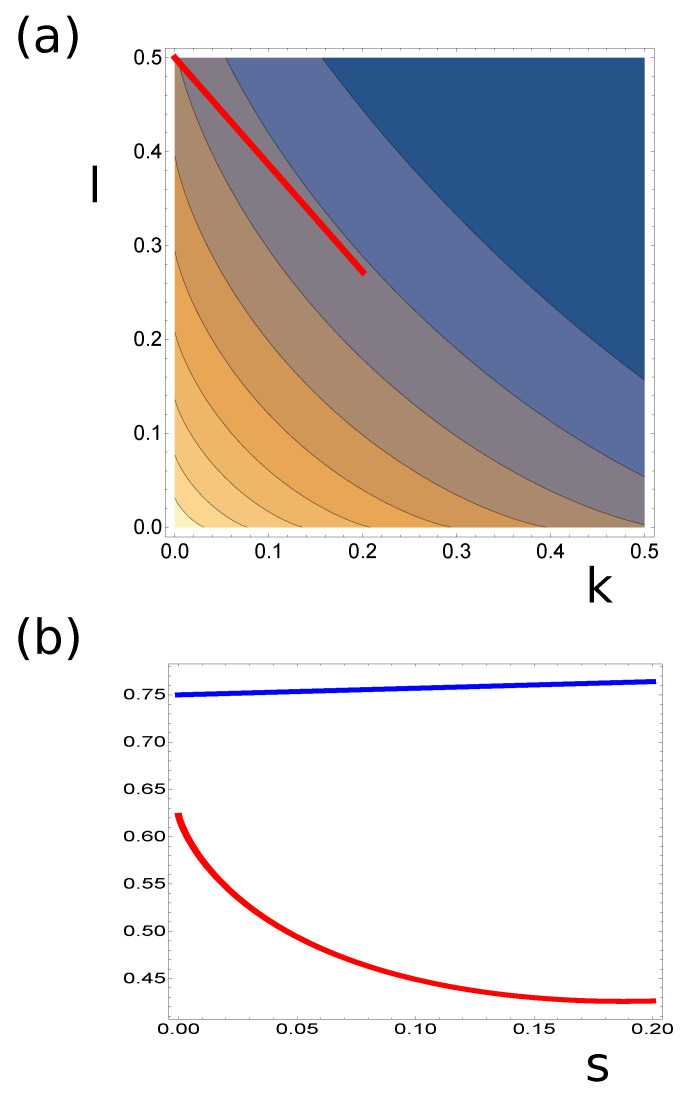
(**a**) The countour plot of the mutual information illustrated in Figure 7b with a hypothetical optimization path shown in red. (**b**) The accuracy (blue curve) and the double of mutual information (red curve) along the path marked in (**a**).The value s=0 describes the point (0,0.5).

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
