# Peer review of "Applications of Information Theory Methods for Evolutionary Optimization of Chemical Computers"

_entropy, 2020, doi:10.3390/e22030313_

Round 1

Reviewer 1 Report

The manuscript Applications of information theory methods for evolutionary optimization of chemical computers by J. Gorecki describes a top-down approach to construction of an information processing device based on chemical computing with the use of a chemical reaction possessing excitable and/or oscillator chemical kinetics.

I welcome such an approach, because it provides a method of designing real chemical systems, such as set of coupled chemical reactors on ether an ordinary laboratory scale or a microscale as achieved within microfluidic devices. The information theoretical framework (in this case specifically mutual information and Shannon entropy) in terms of  an evolutionary algorithm that proposes details of such a device based on its assumed function is an idea definitely of interest for a broad readership of those interested in information processing, not only for the community interested in chemical nonlinear dynamics. As such I fully recommend the manuscript for publishing in Entropy.

After reading through the manuscript, I consider its content being either publishable as is, or with minor corrections of style. I suggest one such correction to distinguish the part of text in the introduction, which defines the general algorithm for dataset classification, as a separate theoretical section. In particular, the text starting with ‘Within the top-down approach…’ on page 3 and ending on page 4, prior the section Results. This new section may tentatively be called Model for dataset classification. This would split the current section Introduction to an introduction proper and a formulation of the general mathematical model. Another minor correction may replace the phrase word ‘like’ with ‘as’ or ‘such as’ which seems to me more appropriate for a formal mathematically oriented text.

Author Response

I am grateful to the referee for his comments.

The manuscript has been modified and the definition of general algorithm for database classification has been moved to the section Results.

The suggested language corrections have been introduced.

Reviewer 2 Report

This is a nice manuscript that, in my opinion, ultimately deserves publication in "Entropy." I just have a few urgent suggestions to improve the presentation.

The Belousov-Zhabotinsky reaction is an oscillating chemical reaction. In the manuscript the behavior of networks of coupled Belousov-Zhabotinsky oscillators is worked out. Figure 2 of the manuscript shows a Belousov-Zhabotinsky ``building block'': the excitation is taken to last 1 second and the subsequent refractory period is taken to last 10 seconds. During the refractory period no excitation can occur. Subsequent to the refractory period, there is a responsive period of 19 seconds. During that time, the system will excite when it receives a trigger. If no such trigger occurs, the excitation will occur by itself at the end of the responsive period. The similarity between a Belousov-Zhabotinsky oscillator and a nerve cell is obvious at this point. The manuscript starts with an explanation of how a network of coupled reservoirs can be a kind of neural network computer; the oscillators are coupled and an excitation in one oscillator can trigger an excitation in an oscillator that it is coupled to. The oscillatory dynamics of the Belousov-Zhabotinsky reaction can be brought to a temporary standstill by shining light on the reservoir. It is through shining lights on oscillators that information can be input into the system. It is shown in the manuscript how the system can "learn''; through trial and error, the system can adjust parameters and get better at its task. This is significant for an understanding of evolution.

A problem arises when the author of the manuscript develops a mathematical formalism for the functioning of the ``computer'' that he studies. There are many formulae and many symbols. The explanation is not well organized. The system that is ultimately studied has just three coupled oscillators and the author shows how this system can take two input numbers (x and y) and determine which of the two is the larger number. The set-theoretical formalism that is being developed for the coupled Belousov-Zhabotinsky oscillators is, in principle, able to handle much larger sets of coupled oscillators doing much more complicated things. My recommendation is to rewrite much of the manuscript and show a simple derivation that is tailored to show how just this 3-oscillator ``computer'' is able to compare two numbers. As it is, there are too many symbols relating to one another in a much too complicated way. There is t_start, t_end, t_max, t_illum, and t_(p^s). Furthermore, t_start is always larger than t_end?!?! It is still unclear to me what N_test, p_j, and a^s are. People will get lost in this plethora of symbols and set structures. And that is a pity.

The English in the manuscript is generally good. There are a few things. It is somewhat unusual to start a scientific article with ``In my opinion ...'' There is on the whole a lot of first person singular in this manuscript and this is something one generally tries to not do. There are many cases where ``than'' and ``then'' are mixed up. At some point ``bios'' is written instead of ``bias.'' Elsewhere I see ``regent'' for ``reagent.'' Sometimes ``form'' is written instead of ``from.'' Figure 5 is referenced in the text before figure 4.

Author Response

I am grateful to the referee for his comments.

The manuscript has been modified so I hope the quality of result presentation increased.

The language errors have been corrected.

I do not agree with suggestion to cut the manuscript and focus on results for 3-oscillator "computer". The idea of the manuscript is to present general results for application of information theory methods for design of chemical classifiers. The method, as discussed, can be applied to any problem that can be formulated as a database classification. The optimized network of oscillators that can tell which of two numbers is larger is just a nice example that the general method works.

The fact that t_{start} is larger than t_{end}  is just the result of optimization. The formula:

illumination time of input oscillator = t_{start} + (t_{end} - t_{start})*p_j

is meaningful regardless if t_{start} > t_{end} or t_{start} < t_{end}.